# Zebrafish Polymerase Theta and human Polymerase Theta: Orthologues with homologous function

Corey Thomas[1], Sydney Green[1], Lily Kimball[2], Isaiah R. Schmidtke[2], Lauren Rothwell[2], Makayla Griffin[2], Ivy Par[1], Sophia Schobel[1], Yayleene Palacio[1], Jamie B. Towle-Weicksel[1], Steven E. Weicksel [2]*

1 Department of Physical Sciences, Rhode Island College, Providence, Rhode Island, United States of America, 2 Department of Biology and Biological Sciences, Bryant University, Smithfield, Rhode Island, United States of America

* sweicksel@bryant.edu

## Abstract

DNA Polymerase Theta (Pol θ) is a conserved an A-family polymerase that plays an essential role in repairing double strand breaks, through micro-homology end joining, and bypassing DNA lesions, through translesion synthesis, to protect genome integrity. Despite its essential role in DNA repair, Pol θ is inherently error-prone. Recently, key loop regions were identified to play an important role in key functions of Pol θ. Here we present a comparative structure-function study of the polymerase domain of zebrafish and human Pol θ. We show that these two proteins share a large amount of sequence and structural homology. Using a classical biochemical approach we observe that zebrafish Pol θ displays behavior characteristic of human Pol θ, including DNA template extension in the presence of different divalent metals, microhomology-mediated end joining, and translesion synthesis. These results will support future studies looking to gain insight into Pol θ function on the basis of evolutionarily conserved features.

## Introduction

A cell's genome is damaged at a nearly continuous rate due to a combination of internal (i.e., reactive oxygen species and cellular processes) and external factors (i.e., ultra-violet radiation, environmental exposure). Left unresolved, DNA damage has the ability to alter cell function through disruption of genomic stability [1]. To preserve genomic integrity a robust system of DNA repair enzymes has evolved an essential role in fixing the genome and protecting the cell from aberrant function. However, while some repair mechanisms faithfully preserve sequence (i.e., homologous recombination) some sacrifice fidelity and generate mutations (i.e., nonhomologous end joining) to keep the genome together. Despite this mutagenic behavior, these mechanism help avoid cell death due to genomic disfunction due to fragmentation. Unsurprisingly, this ability of DNA repair enzymes to avoid cell death while also driving mutagenesis has made DNA repair enzymes key factors in carcinogenesis. To date, every known DNA repair polymerase and many of their cofactors have been linked to cancer [2–4]. This has made understanding how DNA repair polymerases work and the factors that regulate their activity of great interest.

**Data availability statement:** All relevant data are within the manuscript and its Supporting Information files.

**Funding:** SEW - P20GM103430 - Rhode Island Institutional Development Award (IDeA) Network of Biomedical Research Excellence. - https://web.uri.edu/riinbre/ - RI-INBRE was not part of the study JBTW - R15GM144903-01 - National Institute of General Medical Sciences of the National Institutes of Health - https://www.nigms.nih.gov/ - NIGMS was not part of the study The funders had no role in study design, data collection and analysis, decision to publish, or preparation of the manuscript.

**Competing interests:** No authors have competing interests: Cory Thomas Sidney Green Lily Kimball Isaiah Schmidtke Lauren Rothwell Makayla Griffin Ivy Par Sophia Schobel Yayleene Palacio Jamie B Towle-Weicksel Steven E Weicksel

One emerging factor identified in a recent study of patient derived melanoma samples is DNA polymerase-theta (Pol θ or POLQ) [5]. An A-family DNA repair enzyme, Pol θ is essential for cell function and organismal development [6]. Inherently error prone Pol θ [7,8] plays a predominant role repairing double strand breaks (DSB) in the DNA strand through microhomology-mediated end joining (MMEJ, also known as theta-mediated end joining, (TMEJ)), and translesion nucleotide bypass [9–12]. Unlike homologous recombination (HR), the favored DSB repair pathway, TMEJ is highly error prone and is proposed to be activated when HR is overwhelmed (when the genome occurs many double strand breaks) and/or inactive (such as in cancer states). The activity of Pol θ in translesion nucleotide bypass plays a critical role in replication, that while perpetuating genomic mutations, allows replicative DNA polymerases to continue replication while also avoiding more DNA DSBs and potential mutagenesis through replication fork collapse [13]. Together this indicates that Pol θ function is intrinsically mutagenic yet required for cell function. This duality, mutagenic enzymatic behavior while also supporting cell survival [6,14,15], along with aberrant Pol θ activity in cancer cells [16–18] that has led to many hypothesizing that Pol θ activity drives carcinogenesis. However, few models for assessing the function of Pol θ as well as the outcome of Pol θ function in the context of an organism exist.

Zebrafish (*Danio rerio*) have long been employed to model organism and would represent a powerful tool to better understand the function of Pol θ in the context of an organism. With a high degree of similarly to humans, zebrafish have nearly 70% homology in their genes and 85% in human disease-related genes [19]. Zebrafish have a predicted Pol θ orthologue of 2576 amino acid residues and has been shown to be essential for fixing double strand breaks during embryonic development [6]. However, little is known about the function of zebrafish Pol θ and if it functions similarity to that of its human orthologue.

Here we present the first comparative analysis of structure and function of the purified polymerase domain (PD) fragment of zebrafish POLQ (zPol θ or zPOLQ) and human POLQ (hPol θ or hPOLQ). Protein alignment indicates that many of the residues present in the polymerase domain between the two proteins are conserved resulting in similar folded structures. However, within loop regions (unresolved in the human crystal structure), specific to PolQ relative to other A-type proteins, there is little conservation. Despite this lack of conservation, we observe similar zPOLQ behavior compared to hPOLQ. zPolQ can extend DNA templates even in the presence of conventionally inhibitory Ca$^{2+}$, perform TMEJ, and bypass DNA lesions, hallmarks of PolQ function in the cell.

## Results

### Zebrafish and human polymerase domains display high degree of structural similarity

To determine the degree of similarity between zPol θ and hPol θ proteins the primary amino acid sequences were aligned (Table 1, S1 Fig). The alignment of the full-length Pol θ proteins indicates, zebrafish and human Pol θ share 46% identity. This degree of similarity increases when comparing the predicted polymerase domain (63%), as well as subdomains containing catalytic activity, fingers (75.3%), thumb (74.6%), and palm (66.7%). These data suggest a structurally similar molecule.

To assess the extent of structural similarity we generated a predicted structure for zPol θ PD using ColabFold [20] to compare to the solved crystal structure of hPol θ PD [21]. Upon visual inspection the predicted zPol θ PD displays classical DNA polymerase PD structures (Fig 1). The three major subunits, the fingers, thumb, and palm are visible, and when modeled in, a DNA molecule can fit in the presumed catalytic domain. The model also indicates the

presence of unstructured loop domains, that have functional importance [12], that were not resolved in the hPol θ PD structure. An overlay of the hPol θ PD and zPol θ PD show that the structures have a high degree of similarity. As predicted by the amino-acid alignment, these data indicate that much of the structure of the hPol θ PD is conserved in zPol θ.

## hPol θ and zPol θ PD are structurally similar

The plasmid containing the c-terminal polymerase domain fragment of recombinant zPol θ (residues 1832–2576) was expressed and purified same as hPol θ PD (18, summarized in the Materials and Methods). Similar to hPol θ PD, one protein preparation of zPol θ PD yields approximately 5–10 µM of protein, with similar expression and purification levels observed (Fig 2A)

To confirm similarity in secondary structure between hPol θ PD and zPol θ PD, circular dichroism spectroscopy (CD) was performed at 20°C. The same sample was heated from 20–90°C in order to determine the thermal denaturation profile. Both spectra were overlayed

Table 1. Polymerase Theta Sequence alignment analysis.

| | Zebrafish | | Human | | % Identity |
|---|---|---|---|---|---|
| Full Length protein | 2576 aa | | 2590 aa | | 46% |
| Polymerase Domain | 744 aa | 1832-2576 | 799 aa | 1791-2590 | 63% |
| Thumb Sub-domain | 122 aa | 2093-2217 | 125 aa | 2093-2217 | 74.6% |
| Palm Sub-domain | 163 aa | 2217-2252; 2469-2576 | 177 aa | 2218-2253; 2314-2332; 2475-2590 | 66.7% |
| Fingers Sub-domain | 142 aa | 2327-2468 | 142 aa | 2333-2474 | 75.3% |

*__TT__ represents site of CPD lesion

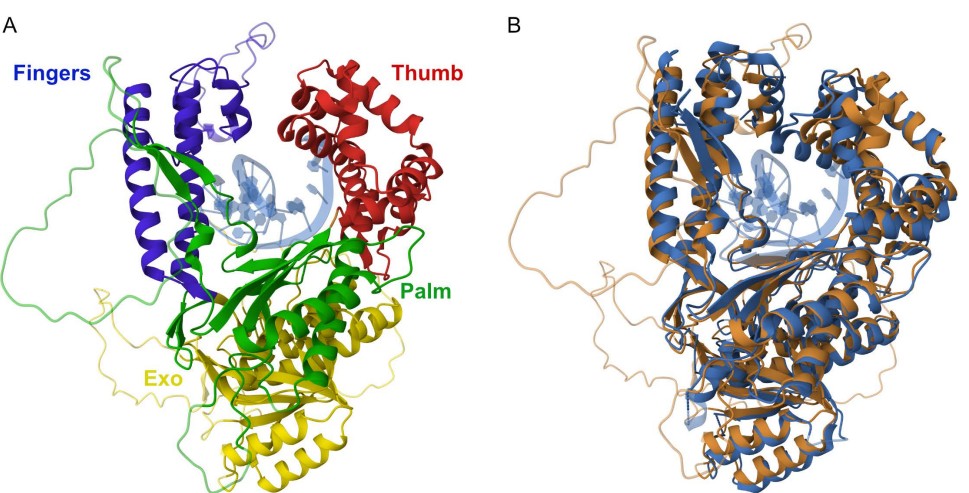

**Fig 1. Structural modeling of zPol θ PD. (A)** AlphaFold rendering of zPol θ PD with subdomains colored, thumb residues 2093-2217 – blue, fingers residues 2327-2468 – red, palm – residues 2217-2252; 2469-2576 green, and exo-nuclease residues 1832-2093 – yellow. DNA is colored in light blue. **(B)** FATCAT overlay of AlphaFold rending of zPol θ PD (bronze) and hPol θ PD (navy) crystal structure [21] without loop inserts. DNA is colored in light blue. PDB 4X0Q.

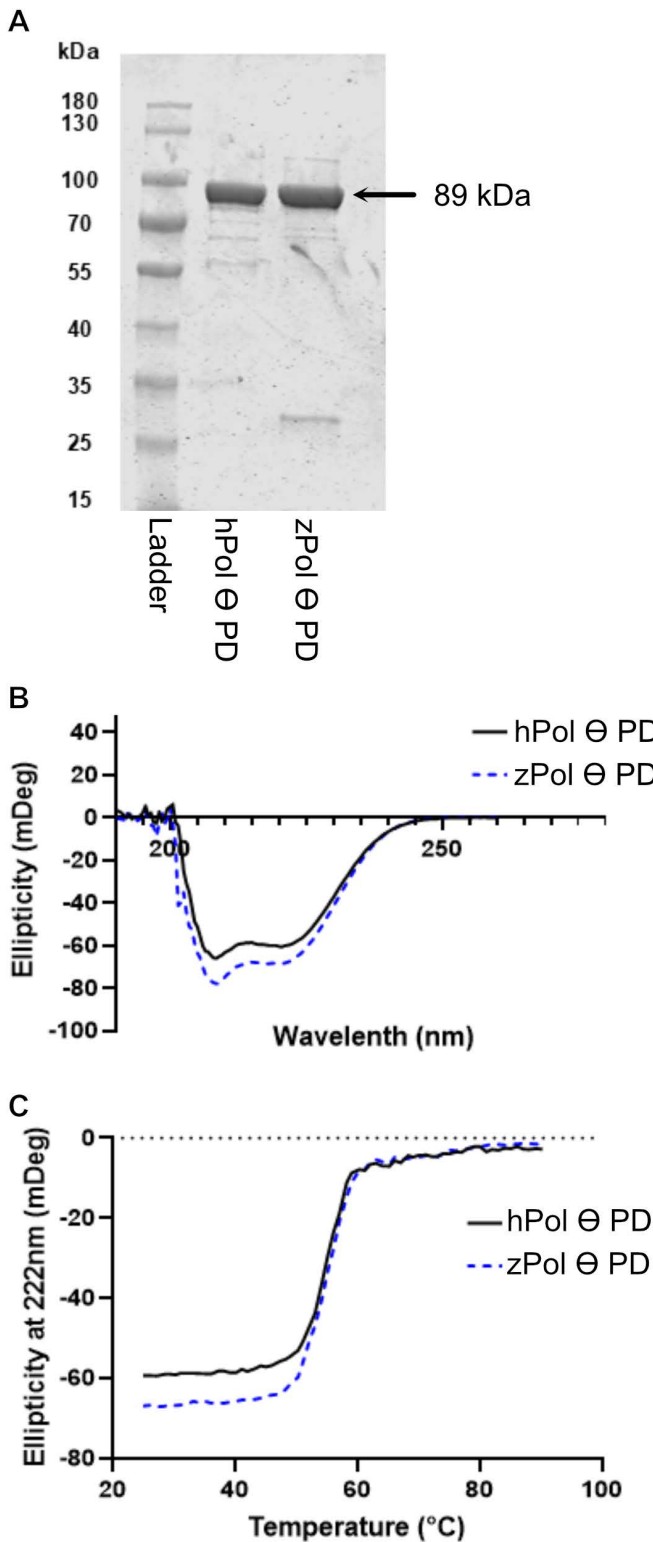

**Fig 2. hPol θ and zPol θ PDs display similar secondary characteristics and thermal stability.** (A) SDS-PAGE of purified hPol θ PD and zPol θ PD. Expression and purification of zPol θ PD was the same as hPol θ PD as described in the Materials and Methods. For each sample, approximately 56–60 pmol of cleaved, purified protein were loaded on a 10% SDS PAGE and Coomassie stained. Both hPol θ and zPol θ PD fragments migrate to approximately 90 kDa as expected. (B) Circular dichroism spectra of 3 μM hPol θ PD (solid line) and zPol θ PD (dashed) proteins in 10 mM Potassium Phosphate buffer. Samples were scanned from 190 to 280 nm. (C) The same samples were heated from 20–90°C and ellipticity measured at 222 nm.

and indicated minimal variance suggesting that both hPol θ and zPol θ PD have similar secondary characteristics and thermal stability with a Tm of approximately 55°C.

## zPol θ PD binds to dsDNA substrate

DNA binding by a DNA polymerase is one of the first steps in its catalytic mechanism. To determine the DNA binding capabilities of zPol θ PD, we titrated zPol θ PD from 0–1000 nM protein against 10 nM 25/40 dsDNA (Fig 3). Complexed DNA/protein products were separated on a non-denaturing gel to determine a dissociation constant ($K_{D(DNA)}$) for DNA binding. Similar to hPol θ PD, zPol θ PD has a low $K_{D(DNA)}$ value of approximately 19.8 ± 3.1 nM.

## zPol θ PD can extend dsDNA similar to hPol θ PD

The second step in the DNA polymerase catalytic pathway is nucleotide binding and formation of the phosphodiester bond. To explore this fundamental step of DNA Polymerase activity, we assayed zPol θ PD's ability to extend 25/40 dsDNA under varying conditions. Under standard steady-state conditions, 50 nM of zPol θ or hPol θ was pre-incubated with 200 nM 25/40 dsDNA. The reaction was initiated by the addition of 125 nM dNTP as described in Fig 4 along with 20 mM $MgCl_2$, the preferred metal for DNA polymerase [22]. We observed under these conditions both hPol θ and zPol θ PDs were able to extend the full 18-mer template with all nucleotides present (Fig 4A). Both enzymes were able to incorporate single nucleotides, correct and incorrect as well. Notably, zPol θ PD was able to incorporate incorrect dGTP to full extension (n+1) compared to only n+6 with hPol θ PD. DNA polymerases can utilize other metals including $Mn^{2+}$, and we observed an increase in mutagenesis through misincorporation for both hPol θ and zPol θ PDs. When provided with all dNTP, zPol θ PD can extend past the end of the template (n+18). Overall, zPol θ PD experiences more extension products especially with incorrect nucleotides dATP, dGTP, and dTTP compared with hPol θ PD under similar conditions. To address if the increased extension products are a result of increased processivity or multiple polymerization events, we repeated the primer extension assay with a 25-fold excess of 5'-FAM labeled dsDNA to Pol θ and challenged extension in the presence of excess unlabeled DNA (Fig 4B). In the presence of the trap DNA, we observed multiple band products for both hPol θ PD and zPol θ PD indicating processivity up until n+6 extension suggesting multiple initiation events for longer products consistent with previous work [12].

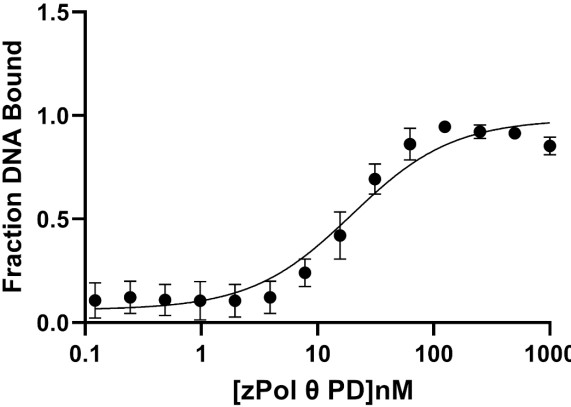

**Fig 3. zPol θ PD binds tightly to dsDNA.** zPol θ PD was titrated from 0–1000 nM against 10 nM 25/40 dsDNA. Bound and unbound products were separated on a 6% non-denaturing gel and quantified using ImageQuant. $K_{D(DNA)}$ was mathematically calculated using Equation 1 and is the midpoint between bound and unbound fractions.

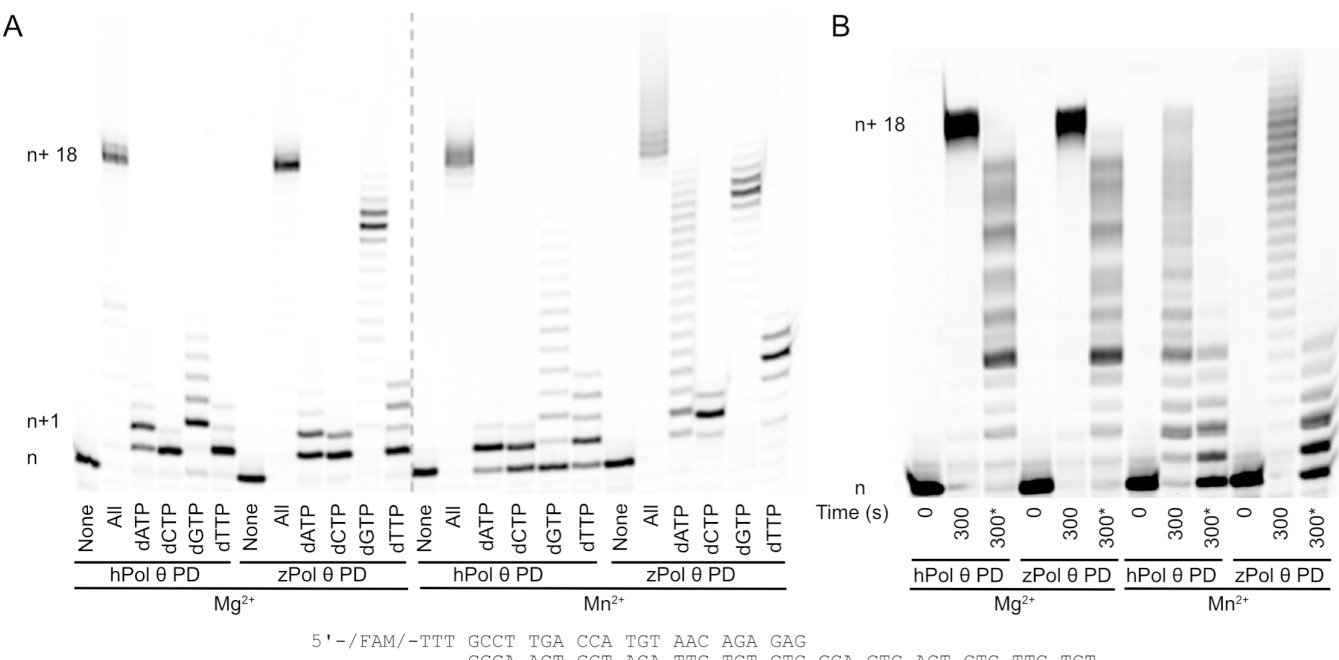

**Fig 4. zPol θ PD experiences greater nucleotide extension activity compared to hPol θ PD.** Denaturing gel showing primer extension dsDNA substrate with MgCl₂ and MnCl₂ (A) Under steady-state conditions 50 nM hPol θ or zPol θ PD proteins were preincubated with 200 nM 25/40 dsDNA and combined with either 10 mM MgCl₂ or MnCl₂ for 5 minutes at 37°C. (B) The assay was repeated under steady-state conditions, with 10 nM Pol θ and 25 nM 25/40 dsDNA. Reactions were initiated with 100 μM dNTPs, 10 mM MgCl₂ or MnCl₂, and 75-fold excess of unlabeled 25/40 dsDNA for 5 minutes at 37°C (indicated by a *). DNA extension products were separated on a denaturing gel and visualized on a Typhoon scanner. Each n+1 band represents an extension of one nucleotide following the DNA template as described above. n+1 would represent either correct nucleotide incorporation of dCTP opposite a templating G (underlined) or a misincorporation event of dATP, dGTP, or dTTP opposite templating G. Each subsequent band is another nucleotide extension with a maximum template-dependent extension of n+18. Bands migrating higher than n+18 represent de novo synthesis.

Steady-state conditions highlight overall DNA polymerase activity, but because the dsDNA substrate is in excess, activity highlights multiple turnovers [23]. Although DNA hPol θ PD has been shown to have robust de novo activity with manganese [24], we wanted to be sure this overextension observed with zPol θ PD and Mn²⁺ was the result of extension and not an artifact. We changed the ratio of protein to DNA to reflect single-turnover conditions; excess protein over dsDNA substrate. Here we are able to observe polymerization events for theoretically every available DNA substrate. Similar to steady-state conditions, we observe an even more robust de novo extension with not only all nucleotides, but also with dATP, suggesting that zPol θ PD misincorporation with dATP is preferred (S2 Fig).

## zPol θ PD catalytic activity similar to other DNA polymerases

To further explore the mechanism of nucleotide incorporation of zPol θ, we assayed zPol θ PD under presteady-state conditions in which there is an excess DNA substrate to enzyme with correct nucleotide. This assay focuses on the DNA polymerase ability to extend DNA by incorporating the correct nucleotide opposite a templating base. This activity is biphasic in which there is a rapid polymerization step of nucleotide incorporation at the DNA primer's 3'OH and a slower, rate limiting step of product release [23]. If biphasic activity is not observed, it suggests a step before nucleotide incorporation is the rate-limiting step [25]. To ensure that zPol θ PD follows the traditional DNA polymerase mechanism, 100 nM zPol θ PD was preincubated with 300 nM 25/40 dsDNA. The DNA/Pol θ PD complex was rapidly

combined with 100 μM correct nucleotide and 10 mM $MgCl_2$ from 0.004–0.6 seconds. DNA products were separated on a denaturing polyacrylamide gel and primer extension of n+1 was quantified and data fit to a full biphasic burst equation. zPol θ PD fit to a biphasic equation with an observable polymerization rate ($k_{obs}$) of 15.9 ± 2.5 s⁻¹ (Fig 5).

## zPol θ PD performs MMEJ activity

One of the major functions of DNA Pol θ is its ability to repair double-strand breaks and is the primary DNA polymerase for microhomology-mediated end joining. In doing so, Pol θ utilizes internal homology within the DNA sequence to act as a template. Pol θ aligns these complementary pieces and extends in the 5' to 3' direction [7,26,27]. hPol θ PD has been shown to able to perform MMEJ activity on short 12-mer single-stranded DNA, but the full 290 kDa Pol θ with the N-terminal helicase and central domains are needed to anneal and extend larger segments of DNA [26]. We wanted to ensure that zPol θ PD could also perform MMEJ similar to hPol θ PD on short fragments of DNA. Fig 6 is a representative gel of hPol θ and zPol θ performing MMEJ on a 12-mer ssDNA fragment. As indicated in the schematic above, the CCCGGG are aligned through Pol θ PD in the presence of dNTP and subsequently extended in the opposite direction giving rise to a slower moving double-stranded DNA product. Both hPol θ PD and zPol θ PD are able to perform this activity which we further verified on a denaturing gel (S3 Fig). We hypothesize the smaller product bands are indicative of classic snap-back synthesis in which the DNA substrate anneals onto itself for Pol θ to extend [26–29]. This behavior has been observed by others on hPol θ and there is little variation between the two species [26].

## zPol θ is able to bypass CPD lesion DNA

Pol θ is a versatile DNA polymerase in not only can it perform MMEJ, it has also been shown to bypass cyclobutane pyrimidine dimers (CPD), abasic sites, and 8-oxo-guanine (8-oxoG) lesions [8,12,21,30–32]. By being able to extend a DNA primer passed a template containing a contorted Thymine-Thymine lesion, human and mouse Pol θ have been demonstrated to be critical in suppressing DNA damage and preventing skin lesions [8]. On a molecular level,

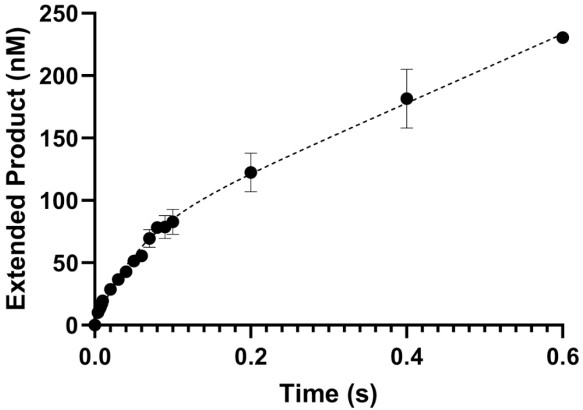

**Fig 5. zPol θ PD experiences biphasic burst activity.** ZPol θ PD (100 nM) was preincubated with 300 nM 25/40 dsDNA. The DNA/Pol θ PD complex was rapidly combined with 100 μM dCTP (correct nucleotide) and 10 mM $MgCl_2$. Reactions were carried out at 37°C and quenched with 0.5 M EDTA. Products were separated on a denaturing gel and quantified with ImageQuant software. Data were fit to a biphasic burst equation to obtain observed $k_{obs}$ rates 15.9 ± s⁻¹. The slower rate $k_{ss}$ was calculated to be 3.4 ± 0.46 s⁻¹.

hPol θ has demonstrated that not only can it insert opposite the initial T in the T-T dimer but is able to mutagenically extend past this lesion for the remaining DNA template. We hypothesized that zPol θ has the same ability to bypass these lesions and we assayed both Pol θ PDs under single-turnover conditions (4:1 protein to DNA) in the presence of damaged DNA

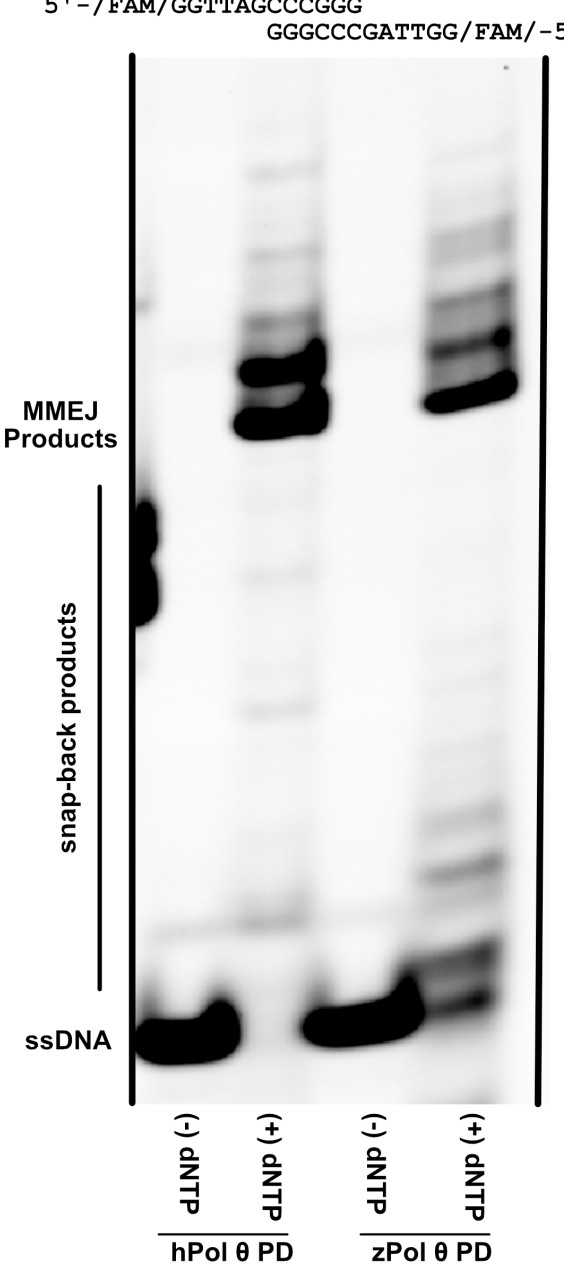

**Fig 6. zPol θ PD is able to perform MMEJ activity of short DNA fragments.** Non-Denaturing gel demonstrating alignment and extension of ssDNA. zPol θ PD (20 nM) was preincubated with 30 nM 5'-FAM 12-mer ssDNA in reaction buffer. Samples either had all nucleotides (+dNTP) or no nucleotides (-dNTP) added and the ternary complex was incubated for 45 minutes at 37°C. Reactions were stopped and products separated on a 12% Native PAGE. The gel was visualized on a Typhoon scanner. MMEJ products are indicated, as well as smaller snap-back products indicated by the bracket.

templates with both $Mg^{2+}$ and $Mn^{2+}$. As predicted, there was little variance in bypass activity of zPol θ PD compared to hPol θ PD on all damaged DNA substrates (Fig 7). Both enzymes were able to readily insert opposite a T-T dimer as well as extend past this lesion with, all dNTPs present, dATP, and to some extent dGTP. Both Pol θ PDs could not incorporate dCTP opposite T-T, but we observed only insertion of dTTP opposite and no extension. Similar incorporations patterns were also observed with the abasic and 8-oxoG lesions. In the presence of $Mn^{2+}$, both Pol θ PDs readily bypassed all damaged DNA, again demonstrating de novo synthesis past the template. zPol θ PD was more robust in extension with the other incorrect nucleotides suggesting $Mn^{2+}$ has an increased mutagenic effect.

### zPol θ PD experiences unusual extension of DNA substrates in the presence of $Ca^{2+}$

To explore the role of divalent metals in DNA polymerase activity for Pol θ, we performed a DNA polymerase extension assay again with hPol θ and zPol θ PDs with their specific DNA substrates swapping out the active metals for $Ca^{2+}$ which has traditionally used as an inert control. Unlike the other divalent metals, $Ca^{2+}$ allows for ternary complex formation, but extension is limited or slow [33,34]. Using 50 nm of the 24/33 undamaged and CPD damaged DNA substrate, we performed a primer extension assay with 200 nM of either hPol θ PD or zPol θ PD with $Mg^{2+}$ substituted for $CaCl_2$. Extension products were separated on a denaturing polyacrylamide gel and quantified based on the percent extension. Fig 8A is a representative gel of extension on 24/33 undamaged DNA template. We observe that hPol θ PD could incorporate nucleotides to some extent, with an n+3 extension product only observed in the presence of all dNTPs and n+1 or n+2 incorporation with purines. zPol θ PD was observed to generate full extension product (94%) on this DNA template (n+12) with all nucleotides present and, like hPol θ PD, could extend with purines as well. Incorporation of dATP led to 92% conversion to product although the enzyme stalled around n+2. Interestingly, zPol θ PD appears to skip the first thymine in the undamaged sequence for both all nucleotides and dATP. The same experiment was carried out with 24/33 CPD damaged DNA. Under these conditions we report that $Ca^{2+}$ reduced DNA polymerase activity for both hPol θ and zPol θ PDs with incorporation of only one nucleotide irrespective if that nucleotide was matched or mismatched with the templating base.

## Discussion

### Zebrafish and human Pol θ structures have a high degree of similarity

Zebrafish serve as a valuable model organism due to their high degree of evolutionary conservation with other metazoans, particularly humans, as well as their relatively low maintenance costs and minimal husbandry requirements compared to other vertebrate models. Prior to our study, several genetic investigations have examined the effects of loss-of-function mutations in DNA polymerases on zebrafish development. Specifically, studies have explored the roles of DNA polymerase beta (Pol β), Pol θ, and DNA polymerase gamma (Pol γ) in DNA repair and the resolution of double-strand breaks that occur during metazoan embryonic development [6,35–38]. Findings from these studies suggest that these DNA polymerases function in conserved pathways similar to their human orthologs. Additionally, one study demonstrated that recombinant zebrafish Pol β exhibits enzymatic activity comparable to that of mammalian (rat) Pol β [39]. These findings further support the functional conservation between zebrafish and human DNA polymerases, reinforcing our hypothesis that zPol θ is the ortholog of hPol θ.

First, we compared the amino acid sequences (Table 1) and structures (Fig 1) of zPol θ and hPol θ revealing that the two proteins share a high degree of similarity. Importantly, the

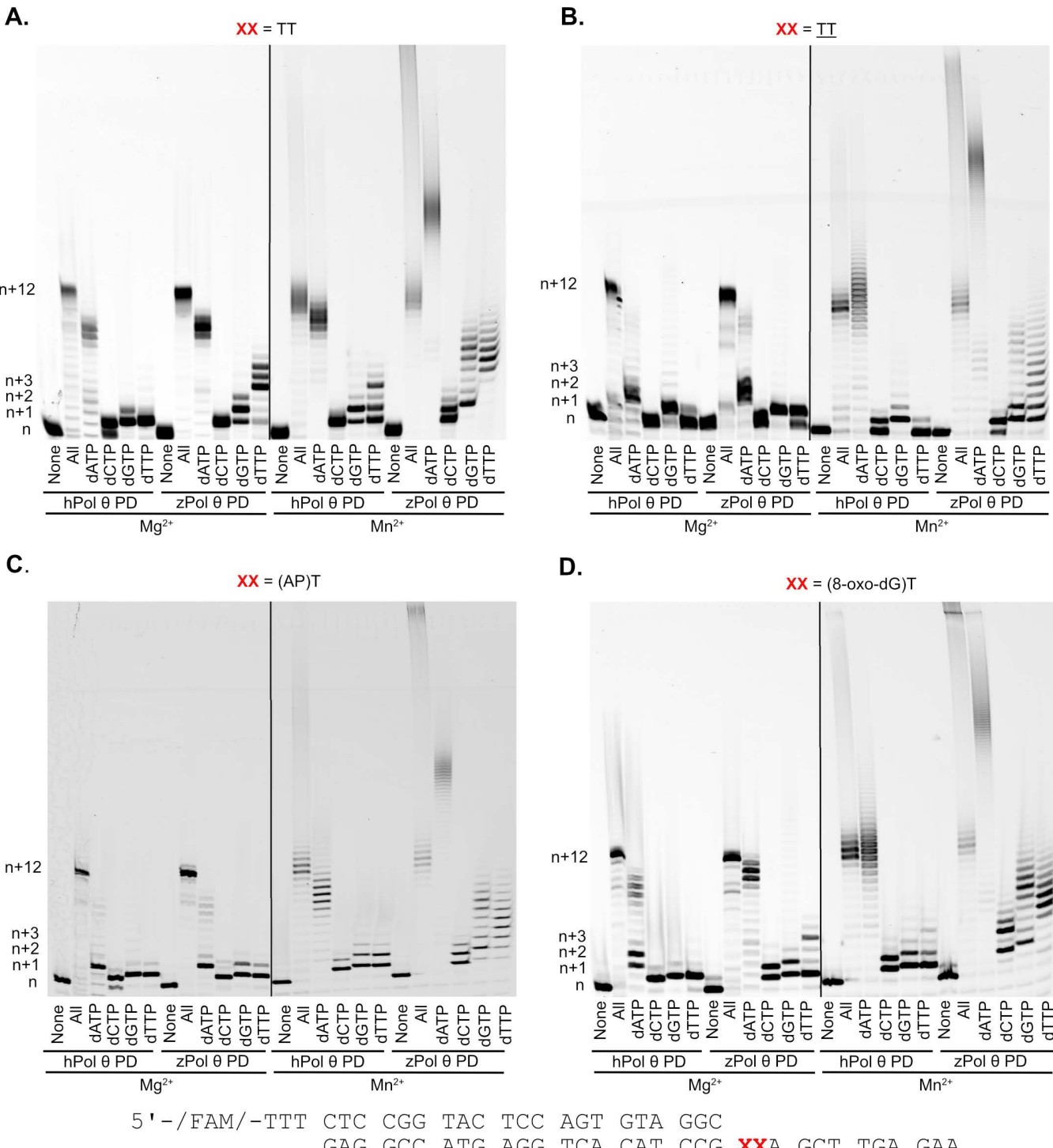

5'-/FAM/-TTT CTC CGG TAC TCC AGT GTA GGC
         GAG GCC ATG AGG TCA CAT CCG **XX**A GCT TGA GAA

**Fig 7. zPol θ PD is able to bypass DNA lesions.** Denaturing gel demonstrating primer extension on damaged dsDNA substrates with $MgCl_2$ and $MnCl_2$. As described in the Materials and Methods, 200 nM hPol θ and zPol θ PDs were preincubated with 50 nM DNA substrate, (A) Undamaged (TT), (B) CPD (TT), (C) Abasic ((AP)T), (D) 8-oxo-dG ((8-oxo-dGT)T). Reactions were initiated by the addition of 125 nM dNTPs as described and either 10 mM $MgCl_2$ or $MnCl_2$. Reactions were carried out at 37°C for 5 minutes and products visualized on a 12% denaturing gel. Higher migrating products are indicative of full extension (n+12) with smeared bands representing de novo synthesis with extension past n+12.

**A**

```
5'-/FAM/-TTTCTCCGGTACTCCAGTGTAGGC
        GAGGCCATGAGGTCACATCCGTTAGCTTGAGAA
```

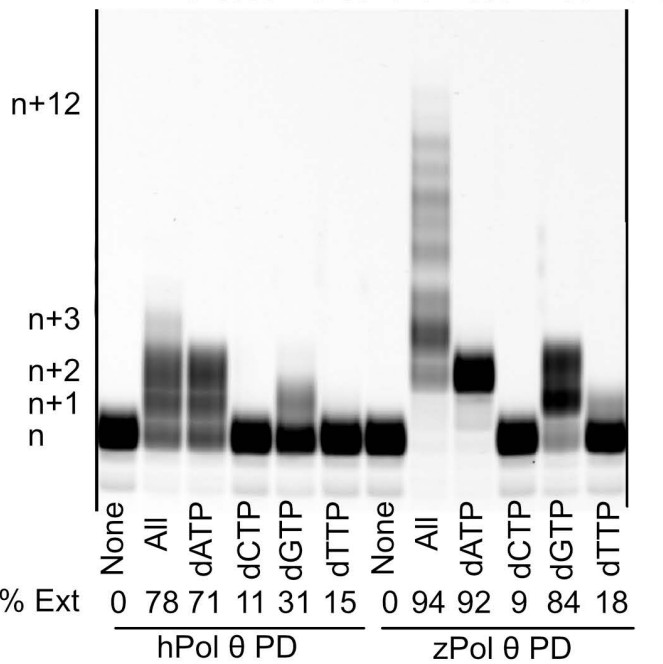

**B.**

```
5'-/FAM/-TTTCTCCGGTACTCCAGTGTAGGC
        GAGGCCATGAGGTCACATCCGTTAGCTTGAGAA
```

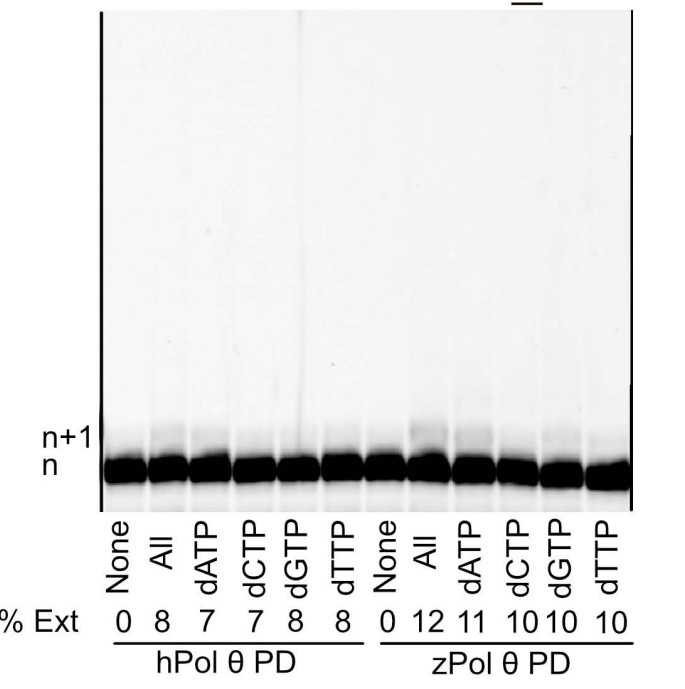

**Fig 8. DNA Pol θ PD is able incorporate and extend dsDNA in the presence of Ca²⁺.** Denaturing gel showing primer extension dsDNA substrate with CaCl$_2$. Under single turnover conditions 200 nM Pol θ PD (human or zebrafish) was preincubated with 50 nM 24/33 undamaged (A) or damaged (**B**) DNA substrate and reacted with 125 nM nucleotides in 10 mM CaCl$_2$. Reactions were carried out at 37°C for 5 minutes and products visualized on a 12% denaturing gel. Percent extended was calculated using ImageQuant software by quantifying the intensity of the extended products (n+1 and higher) divided by the intensity of the total amount of DNA.

areas of greatest similarity are in the catalytic subdomains (fingers, thumb, and palm) of the polymerase domain. This is not surprising since the polymerase domain would be the most likely to retain its homology as to retain its function across evolutionary time. Interestingly, unlike other A-type polymerase family members Pol θ has three loop structures within the PD which have been identified to be important for function [12]. Comparison of the zPol θ and hPol θ sequences indicate that zPol θ also contains these inserted loops however, they contain little homology (S1 Fig) to that observed between human and mouse [21] Pol θ. Despite this difference, expression and purification yields of zPol θ PD were similar to that of hPol θ PD as were the secondary structural characteristics and thermal stability (Fig 2); an early indication similar protein folding. We also show here that zPol θ PD still retains the same activity observed in hPol θ PD.

## zPol θ PD extends dsDNA

The primary role of Pol θ is to perform microhomology-mediated end joining and thus, the major goal of this study was to determine if zPol θ PD retained this function. Initially, we tested basic function shared with all polymerases, asking if zPol θ PD could bind to a DNA template and extend it? Our data suggests that our zPol θ PD fragment can, binding tightly to its substrate (Fig 3) with values similar to those of hPol θ PD [40]. This is expected because loop 1, located in the thumb domain or DNA binding domain, is thought to be involved with contacts to DNA [12] and is a highly conserved loop region between zPol θ and hPol θ PD. We also observed that zPol θ PD can extend a DNA substrate (Fig 4) and we show it has robust activity especially in the presence of $Mn^{2+}$ . While most of the data presented in this work was qualitative, probing how fast a DNA polymerase makes a phosphodiester bound through biochemical kinetics can provide insight into mechanism of incorporation [23]. Like most DNA polymerases, zPol θ PD performs biphasic burst kinetics which is indicative of a two-step mechanism with a rate limiting step of product release. zPol θ PD experiences an observed polymerization rate of around $16\,s^{-1}$ (Fig 5) which is almost 4 times slower than its human ortholog [40], but not uncommonly slow as a similar DNA Polymerase β experiences a similar rate [41–43]. Why zPol θ PD might experience a slower rate is unknown. It could be due to the lack of conservation within the looping structures in the palm domain which in human Pol θ may drive substrate alignment for rapid polymerization.

## zPol θ PD is able to perform microhomology-mediated end joining

It is also important that zPol θ PD also be able to perform MMEJ as in hPol θ as it is Pol θ's primary function in the cell. Although a majority of human Pol θ's N-terminal and central domains are critical for MMEJ, studies have shown that the PD of Pol θ retains limited function for aligning and extending short single-stranded DNA [26]. zPol θ PD was no exception (Fig 6) and was able to complement two single-strand DNAs and extend. Providing perhaps the most compelling evidence of homologous function between the human and zebrafish Pol θ PDs.

## zPol θ PD can bypass CPD lesions

Another function of hPol θ is its ability to bypass DNA damage. A recent study in mice suggest that bypassing UV damage is critical in the prevention of skin cancer [8]. Our data provides evidence that zPol θ PD is able to bypass and extend past CPD lesions *in vivo* similar to that of hPol θ PD (Fig 7). Translesion bypass activity has been highlighted as a function of loops 2 and 3 in the hPol θ PD [12,21]. Surprisingly, zebrafish display very little homology through similar inserts (S1 Fig). Our structural analysis might suggest that the critical

residues for this function may be located at the c-terminal end of this insert beginning with the sequence GMXFSXSMR, as this is the most highly conserved portion of the loop. Further studies exploring this insert in zebrafish are needed to determine if these conserved amino acids are truly critical. Then it can be determined if the function is dictated by the overall presence of the loop or that the loop dependent activities require the few conserved amino acids retained in zPol θ.

### zPol θ PD experiences Ca$^{2+}$ dependent polymerization

Our data shows that zPol θ PD retains all of the DNA polymerase activities of hPol θ PD, with the exception being, a robust Ca$^{2+}$ dependency during DNA extension (Fig 8). While it is unusual to see DNA polymerases extend past the initial insertion event with calcium, there have been instances where high-fidelity *Sulfolobus solfataricus* Dpo4 polymerase uses calcium [44]. Whether or not swapping Ca$^{2+}$ in zebrafish makes it a faster or mutagenic polymerase has not been explored but evolutionarily it is possible that Ca$^{2+}$ makes for a stable ion swap for structural alignment and catalytic activity. Given that in the freshwater habitats of zebrafish likely contain nearly identical concentrations of the divalent ions [45], its plausible that Ca$^{2+}$ could readily be a co-factor substitute.

Our data presented is clear evidence that zPol θ is a homolog to hPol θ and that structurally and enzymatically they have similar functions. This study is significant as it highlights zebrafish as a model organism for studying Pol θ and its potential function in DNA repair and disease. Given the robust assortment of tools, zebrafish offers a powerful, functionally relevant model for human melanoma. Future studies could introduce patient derived mutations by way of germline alterations and study the effects over the lifetime of the animal. Thus, adding new insights into potential disease markers and mechanisms of disease progression and treatment.

## Materials and methods

### Materials

All materials were purchased from Sigma-Aldrich (St. Louis, MO), Bio-rad Laboratories (Hercules, CA), AmericanBio (Canton, MA), and Research Products (Mount Prospect, IL). DNA oligonucleotides were purchased from Integrated DNA Technologies (Newark, NJ) and deoxynucleotides from New England Biolabs (Ipswich, MA). All DNA oligos were purified via HPLC with standard desalting from the manufacturer.

### Zebrafish Pol θ PD cloning

Total RNA from 4-hour post fertilization embryos was extracted using TRIzol (Invitrogen) following manufactures instructions. A library of cDNAs was generated from the pool of polyA mRNAs using ProtoScript II Reverse Transcriptase (New England Biolabs, NEB) following manufactures instructions, primed by oligo(dT). The polymerase domain of zebrafish Pol θ was then amplified for cloning into the POLQM1 vector [12] a pSUMO3 based expression vector. This was a two-step cloning process as the POLQM1 vector did not have multiple cloning sites.

First, the polymerase domain of zebrafish Pol θ (residues 1832–2576) was amplified from the cDNA library with primers containing a 5' KpnI site and a 3' BamHI site:

Pol θ RVS BamHI- TATACTGGATCCTTATATGTCCAGGTCTTGAAGGTTACC

Pol θ FWD KpnI- ATTAGGTACCTCAACATCAGTGTTAGGCGCAC

Second, the 6xHIS and SUMO sequences (HIS-SUMO) of POLQM1 were amplified off of the plasmid using primers containing a 5' XbaI site and a 3' KpnI site:

HIS-SUMO RVS KpnI- ATTAGGTACCTCCCGTCTGCTGC

HIS-SUMO FWD XbaI- TTCCCCTCTAGAAATAATTTTGTTTAACTTTAAGAAG

PCRs reactions used Phusion High-Fidelity DNA Polymerase (NEB) following manufacturer's instructions and were run for 30 cycles. PCR Products were gel isolated from a 1% TAE agarose gel using Freeze 'N Squeeze DNA gel extraction columns (Bio-Rad), following manufacturer's instructions.

Next, the zPol θ, HIS-SUMO, and POLQM1 DNAs were digested with appropriate enzymes (NEB) overnight at 37°C:

zPol θ PCR - BamHI and KpnI

HIS-SUMO PCR - KpnI and XbaI

POLQM1 - XbaI and BamHI

Digested samples were gel separated on a 1% TAE agarose gel and fragments were isolated using Freeze 'N Squeeze DNA gel extraction columns. zPol θ and HIS-SUMO digested fragments were then ligated using T4 ligase (NEB) incubating at 16°C overnight and gel isolated from a 1% TAE agarose gel using Freeze 'N Squeeze DNA gel extraction columns. zPol θ-HIS-SUMO fragment was ligated into linearized POLQM1 vector using T4 ligase and incubating at 16°C overnight. NEB 5-alpha competent *E.coli* (NEB) were transformed by ligated productions using manufacturer's instructions. Bacteria were selected for ampicillin resistance.

## zPol θ polymerase domain modeling and alignments

Amino acid sequence alignments were completed using EMBL-EBI Clustal Omega MSA [46] on default settings. Structural rendering of the zPol θ polymerase domain was completed using ColabFold v1.5.5: AlphaFold2 using MMseqs2 [20,47–50] on default settings. The resulting zPol θ structure was compared to the solved hPol θ polymerase domain structure (AX0Q) [21] using the pairwise alignment tool in FATCAT [51] on default setting, and visualized using RCSB PDB visualization tools [52].

## Expression and purification of hPol θ and zPol θ

Recombinant pSUMO3 plasmids containing the truncated polymerase domain of hPol θ and zPol θ gene were expressed in *E.coli* and purified as previously described in Thomas et al [40]. Briefly, Rosetta2(DE3) cells containing the zPol θ plasmid were inoculated into autoinduction Terrific Broth and grown at 20°C for 60 hours. Cells were harvested through centrifugation and lysed in Lysis buffer. After 6 rounds of sonication, cell fractions were further separated via centrifugation. The fraction containing soluble protein were applied to a 5mL His-Trap FF crude Nickel Column (Cytiva) by FPLC using a high imidazole gradient. Fractions containing Pol θ were separated again on a HiTrap Heparin HP (Cytiva) column for further purification. Eluate containing Pol θ was incubated overnight at 4°C with SUMO2 Protease (Fisher Scientific) to remove the 6xHIS-SUMO tag. Untagged Pol θ was separated from the 6X-HIS-SUMO on a Hi-Trap Chelating HP column, reserving the flowthrough that contained only Pol θ. A final HiTrap Heparin column removed any remaining non-specific binding proteins and exchanged the imidazole buffer for a high NaCl buffer. Protein purification was verified

on 10% denaturing SDS visualized on an Odyssey CL-x IR scanner (LiCOR). Purified protein is highly unstable and was flash-frozen in liquid nitrogen and stored at -80°C for 3 months maximum.

## DNA substrate generation

Double stranded DNA substrates (dsDNA) were generated using complementary oligodeoxy-nucleotides from IDT. Damaged DNA templates include an abasic site (5' - AAG AGT TCG ATX GCC TAC ACT GGA GTA CCG GAG where **X** indicates an abasic site) and 8-oxoguanine substrate (5' - AAG AGT TCG ATX GCC TAC ACT GGA GTA CCG GAG where **X** indicates 8-oxoG) were also purchased from IDT. Templates representing cyclobutane pyrimidine Thymine-Thymine dimer (CPD) damaged and undamaged were synthesized in the Delaney laboratory (Sarah Delaney, Brown University).

The CPD-containing sequence is as follows: 5'-AAG AGT TCG AXX GCC TAC ACT GGA GTA CCG GAG-3' where XX denotes the CPD lesion. The oligonucleotide was synthesized on a MerMade 4 (BioAutomation) using standard phosphoramidite chemistry. All reagents were purchased from Glen Research. The 5'-dimethoxytrityl group was retained for HPLC purification (Agilent PLRP-S column, 250 mm × 4.6 mm; mobile phase A = 1% acetonitrile, 10% triethylammonium acetate (TEAA), 89% water; mobile phase B = 10% TEAA, 90% acetonitrile). The gradient was as follows: 95% A/ 5% B to 65% A/ 35% B over 35 min at 1 mL/min. Oligonucleotide was subject to detritylation by incubation for 60 min at room temperature in 20% (v/v) aqueous glacial acetic acid. The reaction was quenched by precipitation of the oligonucleotides in room-temperature ethanol. A second HPLC purification was then performed (same column and mobile phases as above) using the following gradient: 100% A/ 0% B to 75% A/ 25% B over 40 min at 1 mL/min. The purified oligonucleotide was flash-frozen with liquid nitrogen and lyophilized.

The 5'6-FAM primers were annealed to complementary DNA templates with sequence context as previously described [8,53] and are described below:

| Substrate | DNA sequence |
| --- | --- |
| 25/40 undamaged | 5'-/FAM/ TTTGCCTTGACCATGTAACAGAGAG CGGAACTGGTACATTGTCTCTCGCACT CACTCTCTTCTCT-5' |
| Unlabeled 25/40 undamaged | 5'-TTTGCCTTGACCATGTAACAGAGAG CGGAACTGGTACATTGTCTCTCGCACT CACTCTCTTCTCT-5' |
| 24/33 CPD damaged | 5'-/FAM/-TTTCTCCGGTACTCCAGTGTAGGC GAGGCCATGAGGTCACATCCGTTAGCTT GAGAA-5' |
| Abasic | 5'-/FAM/-TTTCTCCGGTACTCCAGTGTAGGC GAGGCCATGAGGTCACATCCGXTAGCTTGAGAA – 5' |
| 8-OxoG | 5'-/FAM/-TTTCTCCGGTACTCCAGTGTAGGC GAGGCCATGAGGTCACATCCGXTAGCTTGAGAA – 5' |
| 24/33 undamaged | 5'-/FAM/-TTTCTCCGGTACTCCAGTGTAGGC GAGGCCATGAGGTCACATCCGTTAGCTT GAGAA-5' |
| SS DNA | 5'-/FAM/ GGTTAGCCCGGG |

Confirmation of annealed substrates was determined 12% Native PAGE and samples scanned on an RB Amersham Typhoon Fluorescent Imager (Cytiva) with a FAM filter.

Single oligodeoxynucleotides were purchased from IDT for MMEJ with internal consensus sequence as previously described [26].

## Circular dichroism and melting temperature

Secondary protein characteristics of hPol θ to zPol θ were determined on a J-815-CD Spectropolarimeter (Jasco, Brown University) with a 0.2 cm quartz cuvette at room temperature (20°C). For each sample, 3 μM of protein in 10 mM $K_2HPO_4$ buffer were scanned in triplicate from 190–280 nm. The thermal denaturation profile was determined by using the same instrument by heating the same sample from 20–90°C with a 5°C/min temperature rate increase at 222 nm using the same. Data were analyzed on Prism 10 GraphPad and the melting temperature (Tm) was estimated using the halfway point of the denaturing curve. The assay was repeated twice on two independent protein preparations.

## Electrophoretic mobility shift assay (EMSA)

The DNA binding affinity constant $K_{D(DNA)}$ was determined as previously described [40]. zPol θ was titrated from 0–1000 nM against 10nM 25/40 dsDNA substrate in binding buffer and incubated for 1 hour at room temperature. Samples were separated on a 6% Native PAGE and scanned on an RB Typhoon scanner (Cytiva) with the FAM fluorescence filter. Separated bound and unbound products were quantified using ImageQuant. $K_{D(DNA)}$ was determined by equation 1.

$$Y = \left[\frac{(mx)}{(x + K_D)}\right] + b \tag{1}$$

Here, Y is the ratio of bound protein, m is a scaling factor, x is the concentration of protein, and b is the minimum value of Y. Four replicates and two protein preparations were used to generate this data.

## Rapid chemical quench assay

Biphasic burst kinetics were measured as previously described [40]. Briefly,100 nM Pol θ was pre-mixed with 300 nM 25/40 dsDNA substrate and rapidly mixed with 100 μM of dCTP (correct nucleotide) with 10 mM $MgCl_2$ using an RQF-3 Rapid Chemical Quench instrument (KinTek Corporation) at 37°C between 0.004–0.6 seconds. Reactions were quenched by addition of 0.5M EDTA. Products were separated on a 15% Urea-denaturing polyacrylamide gel and scanned using an Amersham Typhoon RB Fluorescent imager (Cytiva). Extended product (n+1) was quantified using ImageQuant software and then plotted to a full biphasic pre-steady state burst equation via non-linear regression using Prism 9 GraphPad software (equation 2). A minimum of three replicates were included for each assay on two independent protein preparations.

$$[Product] = [E]_{app}\left[\frac{k_{obs}^2}{(k_{obs} + k_{ss})^2} \times \left(1 - e^{-(k_{obs} + k_{ss})t} + \frac{k_{obs}k_{ss}}{(k_{obs} + k_{ss})}t\right)\right] \tag{2}$$

Where $[E]_{app}$ is defined as the burst amplitude, $k_{obs}$ is the observed polymerization rate, and $k_{ss}$ is the steady-state rate.

## Primer extension assays

Qualitative primer extension assays were performed as previously described [40]. Varying conditions were used to explore the primer extension capabilities between the hPol θ and zPol θ. Under single-turnover conditions, excess Pol θ (200 nM) was pre-incubated with 50 nM

DNA and incubated for 5 minutes at 37°C. Nucleotide (125 μM) of either none, all, individual nucleotides were preincubated with 20 mM $x$ inorganic salt (x = MgCl$_2$, CaCl$_2$, MnCl$_2$). Under Michaelis-Menton conditions, DNA (200nM) was in excess to Pol θ (50 nM). All reactions were carried out in buffer containing 20 mM Tris HCl, pH 8.0, 25 mM KCl, 4% glycerol, 1 mM βME, and 80 μg/mL BSA. The reaction was initiated by combining Pol-θ/DNA with dNTP/salt. Reactions were incubated at 37°C for 5 minutes before being stopped by 80% Formamide/EDTA quench. Products were separated out on a 15% urea-denaturing polyacrylamide gel and scanned on an RB Amersham Typhoon fluorescent imager (Cytiva). The assay was repeated at a minimum of three times on two independent protein preparations.

## Processivity assay

Processivity experiments were carried out as previously described [12] and similar to the qualitative primer extension assay described above except 10 nM Pol θ was preincubated with 250 nM 5' FAM labeled 25/40 dsDNA. Unlabeled 25/40 dsDNA was added in excess (18.7 μM) to the mixture containing 100 μM dNTPs and 10 mM of either MgCl$_2$ or MnCl$_2$. Reactions were incubated for 5 minutes. The assay was repeated twice.

## MMEJ assay

Microhomology mediated end-joining assay for both hPol θ and zPol θ were carried out as previously described [26] on a 12-mer FAM labeled oligodeoxynucleotide. Pol θ (20 nM) was preincubated with 30 nM 5'-FAM ssDNA in reaction buffer (25 mM Tris-HCl pH 8.8, 1 mM βME, 0.01% NP-40, 0.1 mg/mL BSA, 10% glycerol, 10 mM MgCl$_2$, 30 mM NaCl) for 5 minutes at 37°C. Nucleotides (20 μM) were added and incubated at 37°C for an additional 45 minutes. Reactions were terminated by addition of non-denaturing stop buffer (100 mM Tris-HCl pH 7.5, 10 mg/mL proteinase K, 80 mM EDTA, and 0.5% SDS) for an additional 15 minutes. DNA products were separated on a 12% native polyacrylamide gel and scanned by an RB fluorescent Amersham Typhoon (Cytiva) with a FAM filter. The assay was repeated at a minimum of three times on two independent protein preparations.

## Gel images

Raw gel images for the gels used in the figures can be found in the supporting information (S1 File).

## Supporting information

**S1 Fig. CLUSTAL O(1.2.4) zPol θ and hPol θ alignment.** Red indicates loop insertions, thumb residues blue, fingers residues red, palm residues green, and exo-nuclease residues yellow.
(PDF)

**S2 Fig. Single-turnover polymerase activity for Pol θ.** Both hPol θ PD and zPolθ PD were assayed under single-turnover conditions at t 4:1 ratio protein:DNA (see Materials and Methods). Pol θ and 25/40 dsDNA were preincubated and combined with either 10 mM MgCl2 or MnCl2 for 5 minutes and 37°C. DNA extension products were separated on a denaturing gel and visualized on a Typhoon scanner.
(PDF)

**S3 Fig. Denaturing gel showing MMEJ products. zPol θ PD (20 nM) was preincubated with 30 nM 5'-FAM 12-mer ssDNA in reaction buffer.** Samples either had all nucleotides

(+ dNTP) or no nucleotides (-dNTP) added and the ternary complex was incubated for 45 minutes at 37°C. Reactions were stopped and products separated on a 12% Native PAGE. The gel was visualized on a Typhoon scanner. Start of MMEJ products are marked with an arrow, smaller snap-back products are indicated by the bracket.
(PDF)

**S1 File. Raw gel images used for figures.** Raw gel images for the gels used in the figures as indicated by the label. "X" indicates a lane not used in the figure, labels correspond to labels in the figure.
(PDF)

## Acknowledgements

We would like to thank Sarah Delaney and Mary Tarantino from Brown University for the generation of the CPD damaged DNA. Thank you to Sylvie Doublié from the University of Vermont for the human pol θ plasmid.

## Author contributions

**Conceptualization:** Jamie B Towle-Weicksel, Steven E Weicksel.

**Formal analysis:** Jamie B Towle-Weicksel, Steven E Weicksel.

**Funding acquisition:** Jamie B Towle-Weicksel, Steven E Weicksel.

**Investigation:** Corey Thomas, Sydney Green, Lily Kimball, Isaiah R Schmidtke, Lauren Rothwell, Makayla Griffin, Ivy Par, Sophia Schobel, Yayleene Palacio.

**Methodology:** Jamie B Towle-Weicksel, Steven E Weicksel.

**Writing – original draft:** Jamie B Towle-Weicksel, Steven E Weicksel.

**Writing – review & editing:** Jamie B Towle-Weicksel, Steven E Weicksel.

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
