## [Decision Letter · Decision Letter 0]

24 Nov 2024

PONE-D-24-49625Zebrafish Polymerase Theta and human Polymerase Theta: orthologues with homologous function.PLOS ONE

Dear Dr. Weicksel,

Thank you for submitting your manuscript to PLOS ONE. After careful consideration, we feel that it has merit but does not fully meet PLOS ONE’s publication criteria as it currently stands. Therefore, we invite you to submit a revised version of the manuscript that addresses the points raised during the review process.Please submit your revised manuscript by Jan 08 2025 11:59PM. If you will need more time than this to complete your revisions, please reply to this message or contact the journal office at plosone@plos.org . Please include the following items when submitting your revised manuscript:

We look forward to receiving your revised manuscript.

Kind regards,

Hari S. Misra, Ph.D.

Academic Editor

PLOS ONE

“SEW

- P20GM103430

- Rhode Island Institutional Development Award (IDeA) Network of Biomedical Research Excellence.

- https://web.uri.edu/riinbre/

- RI-INBRE was not part of the study

JBTW

- R15GM144903-01

- National Institute of General Medical Sciences of the National Institutes of Health

- https://www.nigms.nih.gov/

- NIGMS was not part of the study”

Additional Editor Comments:

Manuscript is reviewed by 2 subject experts and their comments are appended below for your attention. I have gone through it and concur their comments. Additionally, authors are advised to check the processivity of both the polymerases. Results may help to distinguish if length of polymerization shown in Fig 4, is because of higher processivity of the enzyme or the ability of enzyme to reinitiate in multiple events.

Reviewers' comments:

Reviewer's Responses to Questions

**Comments to the Author**

1. Is the manuscript technically sound, and do the data support the conclusions?

Reviewer #1: Partly

Reviewer #2: Yes

2. Has the statistical analysis been performed appropriately and rigorously?

Reviewer #1: No

Reviewer #2: No

3. Have the authors made all data underlying the findings in their manuscript fully available?

Reviewer #1: No

Reviewer #2: No

4. Is the manuscript presented in an intelligible fashion and written in standard English?

Reviewer #1: Yes

Reviewer #2: No

5. Review Comments to the Author

Reviewer #1: In this study by Thomas et al, demonstrated that the polymerase domain of polymerase theta (POLQ) from zebrafish and human has a similar function in terms of amplifying damaged DNA strands. The authors employed several conventional biochemical and biophysical approaches to examine the several functional aspects of human POLQ in zebrafish POLQ. Finally, they conclude stating that both human and zebrafish polymerase domains are functional orthologs. While this is a fundamental investigation and enriches our understanding about the POLQ functionality in a popular biological model organism zebrafish, there are several demerits in the technical design of the study, interpretation of results and providing strong supportive evidence to establish their statement. Hence, the manuscript in its present form does not meet the scientific rigor and publication standard of the concerned journal. The authors may consider the following points to further strengthen the manuscript.

Major concerns:

1) It is not clearly mentioned whether full-length zPOLQ was tested and compared to that hPOLQ.

2) In terms of fold-change, how efficient is the zPOLQ compared to hPOLQ?

3) Regarding the lesion bypass capacity of POLQ, the authors only showed thymidine dimer adduct. Can zPOLQ bypass any bulky adduct, oxidative base damage, or nicked strand lesion in the dsDNA region to extend the single-stranded region?

4) What was the basis of choosing different duplex and SS oligo sequences? Are they preferred binding sequences or any random ones?

5) Regarding the participation of zPOLQ in MMEJ, what was the microhomology sequence length using in the in vitro analysis?

6) An in vitro MMEJ reporter-based assay would be ideal to show zPOLQ's possible functional role in MMEJ pathway.

7) None of the experiments had statistical analysis and no mentioning of the number of biological replicates.

8) Without any in-cell assays, it is difficult to ascertain the functional similarities of these two POLQ domains.

Minor concern:

1) Lacking citations of many of the pioneering reports in the field.

2) The manuscript requires improvement in English writing style.

3) Most images are of poor quality.

4) Figure 1 requires domain marking for easy readout by readers.

5) Figure 2A should indicate the band size and exact domain name of the purified protein. Mention purity level of each of human and zebrafish POLQ domains. Total yield is ambiguous as there are other non-specific bands visible in the gel image.

6) Figure 3A, mention the DNA type used in this assay, preferably in the Y-axis.

7)Figure 6 requires statistical qunatification of the activity.

Reviewer #2: The topic of the manuscript entitled “Zebrafish Polymerase Theta and human Polymerase Theta: orthologues with homologous function” is a detailed and extensive characterization of the Danio rerio (zebrafish) DNA polymerase theta (θ). It is primary research that fills a large gap in our scientific knowledge about an important enzyme of a major model organism. First, the authors show sequence and structural similarity bioinformatically between the human and zebrafish Pol θ (hPol θ and zPol θ, respectively). Then, they overexpress and purify the polymerase domain of both proteins for biochemical characterization. They use a large number of in vitro methods to study and to compare the DNA binding, primer extension, microhomology-mediated end joining and translesion synthesis activities of the two orthologous proteins. They also examine the effect of various divalent cations on the polymerization activity of these Pol θs. The study is well-designed and thorough. Experiments are performed to a high technical standard and are described in sufficient detail. Conclusions are supported by the data. Nevertheless, there are shortcomings in the data presentation that need improvement.

Major Points:

1. Abstract - The authors emphasize the role of loop regions in key functions of Pol θ. However, they do not have any experimental evidence for the role the loop regions of zPol θ play (They have not performed any assays using Pol θ mutants with deleted or point-mutated loops). Therefore, the abstract should be reworded to reflect the true content of the study, and summarize the findings .

2. Fig 1. Structural modeling of zPol θ. - Is this the full length zPol θ or just the polymerase and exonuclease domains? Amino acid numbers should be included in the legend. In Supplemental figure 1 amino acids 1-799 and 1-744 of hPolQ and zPolQ are aligned. However, in Materials and Methods, line 389 says that “polymerase domain of zebrafish Pol θ (residues 1801-2579)” and also elsewhere in the text they refer to “c-terminal zPol θ” (lines 122 and 200). Perhaps a scheme of the domain structure of zebrafish Pol θ showing amino acid numbers should be included in Figure 1. Similarly, in Table 1 exact residues ‘from - to’ should be given.

3. Fig 3. zPol θ binds tightly to ds DNA – Line 151 says that “products were separated on a 6% non-denaturing gel”, however, line 146 contradicts saying “products were separated on a denaturing gel” (probably incorrect, since line 490 states “Native PAGE”). Furthermore, Fig. 3 shows only the quantitation, not the gel. Since the authors declare in Data Availability that “All relevant data are within the manuscript and its Supporting Information files”, gels should be shown too, either in the main Figure 3, or in the Supplementary file. Moreover, since the experiment should be performed three times, error bars should be included in the graph.

4. Fig 4. zPol θ experiences greater nucleotide extension activity compared to hPol θ. - Line 182 states that “Under steady-state conditions 50 nM hPol θ or zPol θ were preincubated with 200 nM 25/40 dsDNA, however line 159 says (probably incorrectly) that “Under standard steady-state conditions, 200 nM of zPol θ or hPol θ was preincubated with 50 nM 25/40 dsDNA”.

5. Fig 5. zPol θ experiences biphasic burst activity. - Again, like for Fig. 3, it would be nice to see the gels that were quantitated here.

6. Fig. 6 zPol θ is able to perform MMEJ activity - This figure is not perspicuous. Firstly, probably a C is missing from the lower strand, and this way it looks as if there is a G:G mispairing with the upper strand. Secondly, it is not clear how many nucleotides Pol θ adds to the ssDNA. Explanations showing ’n’ and ’n+x’ are missing (In the raw gels file there is a caption “double-stranded DNA”). Interestingly, it looks as if a lot of nucleotides are added, at least 10, even though the templating part in the structure is only 6 nt long. Or is the run different just because this is a native gel, not a denaturing one? The authors could pre-anneal the ssDNA for control. Alternatively, the extended product can be run on a denaturing gel to see the exact size or shorter reaction times could be used to see shorter extended products. Line 229-230 in the main text says “We hypothesize the smaller product bands are indicative of classic snap-back synthesis”. This is not clear either, the authors could mark in the figure which “smaller product bands” they mean.

7. Fig. 7 - Unfortunately, Fig. 7 is missing from the manuscript uploaded to PLOS One site. There is another copy of Fig. 4 here. The intended Fig 7 can be judged from the Supplementary raw gels file. The left part of the gel image (with Mg2+) shows some running anomalies and smears. It is not clear if dCTP is incorporated at all, and whether one or two dGTPs are incorporated opposite the T-T dimer. Moreover, it cannot be established after which nucleotide Pol θs stall in case dATP is added. Since probably all experiments have been repeated 3 times, the authors should provide another gel picture with clearer bands.

8. Fig. 8. - Unfortunately, Fig. 8 is also missing from the manuscript! The intended Fig. 8 can be assessed from the Supplementary raw gels file. There are smears/ running anomalies here again. Line 276 says “hPol θ could incorporate every nucleotide to some extent”, however, it seems as if there was no incorporation of dCTP and dTTP. The authors may want to provide another gel picture with clearer bands.

9. Materials and Methods

Template oligonucleotides are given in 3’ to 5’ direction in lines 464, 467 and 470. This contradicts conventions and the format to show dsDNA will not work in text. Annealed ds substrates could be shown in a table, or may not be necessary, if they are shown in the figures (like Fig. 4).

10. Line 494, mx, x, Y, and b should be specified, explained. Similarly, kobs and kss should be specified in line 508.

11. The Discussion needs improvement in writing. E. g. line 314 can be deleted. The authors could discuss whether polymerases of D. rerio have been biochemically or genetically characterized yet.

Minor Points:

1. The manuscript should undergo extensive language and grammar corrections. For example, “zPol θ experiences greater nucleotide extension activity” should read ‘zPol θ exerts/ exercises/ has/ possesses higher nucleotide extension activity’. “zPol θ catalytic activity similar to other DNA polymerases” should read ‘zPol θ catalytic activity is similar to other DNA polymerases’. e.t.c.

2. The References/ Work Cited does not follow a uniform format. The publication date is given as year (e.g. line 610) or year, month (e.g. line 603), or year, month, day (e.g. line 606) or the publication date is missing (line 613).

3. Supplement 2. “hWT and zWT Pol θ” name is used. Are these proteins the same as the ones in the other figures? If so, they should be called ‘hPol θ and zPol θ’, just like in the other figures.

6. PLOS authors have the option to publish the peer review history of their article (what does this mean? ). If published, this will include your full peer review and any attached files.

**Do you want your identity to be public for this peer review?** For information about this choice, including consent withdrawal, please see our Privacy Policy .

Reviewer #1: **Yes: ** Joy Mitra

Reviewer #2: No

---

## [Author Response · Author response to Decision Letter 1]

1 Feb 2025

01/31/2025

Dear Dr. Misra,

We would like to thank the two reviewers and the editorial team for their suggestions for our manuscript. We believe that the changes that we made have made for a much clearer and stronger manuscript for publication in PLOSONE. We believe that this manuscript will be of interest to the readership following markers of melanoma and pol theta dynamics.

We have tracked all changes and below have included our response to reviewer comments below. We thank you for the opportunity to share the work that we have done and the thorough assessment of the manuscript.

Sincerely,

Steven Weicksel

Assistant Professor of Biology

Editorial comments:/

Please include this amended Role of Funder statement in your cover letter; we will change the online submission form on your behalf. We have made the requested changes to our financial disclosure statements, in red.

SEW

- P20GM103430

- Rhode Island Institutional Development Award (IDeA) Network of Biomedical Research Excellence.

- https://web.uri.edu/riinbre/

JBTW

- R15GM144903-01

- National Institute of General Medical Sciences of the National Institutes of Health

- https://www.nigms.nih.gov/

In your cover letter, please note whether your blot/gel image data are in Supporting Information or posted at a public data repository, provide the repository URL if relevant, and provide specific details as to which raw blot/gel images, if any, are not available. Email us at plosone@plos.org if you have any questions. – We double checked and all of our raw gel images are now uploaded in the supporting information file.

Additionally, authors are advised to check the processivity of both the polymerases. Results may help to distinguish if length of polymerization shown in Fig 4, is because of higher processivity of the enzyme or the ability of enzyme to reinitiate in multiple events.

We have included an additional panel to figure 4 addressing processivity (1), determining that both polymerases display similar behavior.

Reviewers' comments:

Reviewer's Responses to Questions

Comments to the Author

1. Is the manuscript technically sound, and do the data support the conclusions?

Reviewer #1: Partly

Reviewer #2: Yes

2. Has the statistical analysis been performed appropriately and rigorously?

Reviewer #1: No

Reviewer #2: No

This is a more qualitative study, but any quantitative work was completed under rigorous biochemical standards with at least two protein preparations and experimentation completed by at least two individuals and noted in the materials and methods. For clarification, we added a statement when a representative gel was used in a figure. Figure legends now include more descriptive language that explains the qualitative nature of the gel.

3. Have the authors made all data underlying the findings in their manuscript fully available?

Reviewer #1: No

Reviewer #2: No

We acknowledge that some raw data was missing. We have made sure that all data was supplied through the supplemental file.

4. Is the manuscript presented in an intelligible fashion and written in standard English?

Reviewer #1: Yes

Reviewer #2: No

We have worked to improve the grammatical errors.

5. Review Comments to the Author

Reviewer #1: In this study by Thomas et al, demonstrated that the polymerase domain of polymerase theta (POLQ) from zebrafish and human has a similar function in terms of amplifying damaged DNA strands. The authors employed several conventional biochemical and biophysical approaches to examine the several functional aspects of human POLQ in zebrafish POLQ. Finally, they conclude stating that both human and zebrafish polymerase domains are functional orthologs. While this is a fundamental investigation and enriches our understanding about the POLQ functionality in a popular biological model organism zebrafish, there are several demerits in the technical design of the study, interpretation of results and providing strong supportive evidence to establish their statement. Hence, the manuscript in its present form does not meet the scientific rigor and publication standard of the concerned journal. The authors may consider the following points to further strengthen the manuscript.

Major concerns:

1) It is not clearly mentioned whether full-length zPOLQ was tested and compared to that hPOLQ. – this was clarified by adding PD to indicate polymerase domain.

2) In terms of fold-change, how efficient is the zPOLQ compared to hPOLQ?- Michaelis-Menten kinetics typically define efficiency by kcat/km, moreover, DNA polymerase kinetics utilize kpol/Kd(dNTP) (2). Those parameters were not defined in this study and are beyond the scope, but are certainly an area of focus for the next manuscript.

3) Regarding the lesion bypass capacity of POLQ, the authors only showed thymidine dimer adduct. Can zPOLQ bypass any bulky adduct, oxidative base damage, or nicked strand lesion in the dsDNA region to extend the single-stranded region? – to address this we have performed additional experiments showing zebrafish Pol θ’s ability to bypass 8-oxo and abasic sites. We have included this Figure 7 of the manuscript as well as the materials and methods.

4) What was the basis of choosing different duplex and SS oligo sequences? Are they preferred binding sequences or any random ones? – Substrate design for the duplex DNA was based on previous referenced studies demonstrating similar function using Pol θ (3). The primer-template DNA is customary for DNA polymerase primer extension assays has been described by previous work (1,3,4). The SS oligo was adapted from previous work exploring TMEJ activity. The SS sequence contains internal consensus sequence (5’-CCCGGG) to facilitate the microhomology alignment and subsequent extension as might see in TMEJ activity (5)

5) Regarding the participation of zPOLQ in MMEJ, what was the microhomology sequence length using in the in vitro analysis? – this was clarified in the document. It was stated at the beginning of the section and in the materials and methods, but we added the substrate length to other occurrences in the text.

6) An in vitro MMEJ reporter-based assay would be ideal to show zPOLQ's possible functional role in MMEJ pathway. – Our approach is an in vitro approach to assess MMEJ function and is well accepted in the field (5,6)

7) None of the experiments had statistical analysis and no mentioning of the number of biological replicates. – These experiments are done in vitro using purified protein. We stated the replicates of preps and number of times the reactions were done. These are all consistent with other previously published studies using purified proteins and our work (3).

8) Without any in-cell assays, it is difficult to ascertain the functional similarities of these two POLQ domains. – The scope of this work is to begin the groundwork for future in-cell characterization. Other studies have shown that Pol θ is required to fix double stranded breaks (7,8). The data presented here provides a solid foundation for further in vivo work.

Minor concern:

1) Lacking citations of many of the pioneering reports in the field. We have added more references, but we are open to suggestions for any other necessary citations as appropriate.

2) The manuscript requires improvement in English writing style. – The text was edited further.

3) Most images are of poor quality. – we reviewed the PLOSONE guidelines, used the suggested app to check for compliance, and made changes to the figures as needed

4) Figure 1 requires domain marking for easy readout by readers. Each domain is identified in the legend with a corresponding color and labeled in the figure. The supplementary figure of the primary sequence alignment has been updated as well to distinguish the amino acids contained in each domain, using the same color coding observed here.

5) Figure 2A should indicate the band size and exact domain name of the purified protein. Mention purity level of each of human and zebrafish POLQ domains. Total yield is ambiguous as there are other non-specific bands visible in the gel image. We have indicated the band size in the figure.

6) Figure 3A, mention the DNA type used in this assay, preferably in the Y-axis. – this is stated in the methods.

7)Figure 6 requires statistical qunatification of the activity. – Figure 6 is a qualitative assay looking for alignment and extension of the ss DNA. Traditionally, this assay is presented this way as in accordance with the literature (5). We have included in the methods that all representative gels have been carried out on at least two protein preparations and from two independent people.

Reviewer #2: The topic of the manuscript entitled “Zebrafish Polymerase Theta and human Polymerase Theta: orthologues with homologous function” is a detailed and extensive characterization of the Danio rerio (zebrafish) DNA polymerase theta (θ). It is primary research that fills a large gap in our scientific knowledge about an important enzyme of a major model organism. First, the authors show sequence and structural similarity bioinformatically between the human and zebrafish Pol θ (hPol θ and zPol θ, respectively). Then, they overexpress and purify the polymerase domain of both proteins for biochemical characterization. They use a large number of in vitro methods to study and to compare the DNA binding, primer extension, microhomology-mediated end joining and translesion synthesis activities of the two orthologous proteins. They also examine the effect of various divalent cations on the polymerization activity of these Pol θs. The study is well-designed and thorough. Experiments are performed to a high technical standard and are described in sufficient detail. Conclusions are supported by the data. Nevertheless, there are shortcomings in the data presentation that need improvement.

Major Points:

1. Abstract - The authors emphasize the role of loop regions in key functions of Pol θ. However, they do not have any experimental evidence for the role the loop regions of zPol θ play (They have not performed any assays using Pol θ mutants with deleted or point-mutated loops). Therefore, the abstract should be reworded to reflect the true content of the study, and summarize the findings . – We have altered the wording.

2. Fig 1. Structural modeling of zPol θ. - Is this the full length zPol θ or just the polymerase and exonuclease domains? Amino acid numbers should be included in the legend. In Supplemental figure 1 amino acids 1-799 and 1-744 of hPolQ and zPolQ are aligned. However, in Materials and Methods, line 389 says that “polymerase domain of zebrafish Pol θ (residues 1801-2579)” and also elsewhere in the text they refer to “c-terminal zPol θ” (lines 122 and 200). Perhaps a scheme of the domain structure of zebrafish Pol θ showing amino acid numbers should be included in Figure 1. Similarly, in Table 1 exact residues ‘from - to’ should be given. – We have altered the text of the abstract to reflect the point of the reviewer. We have also added the residue numbers for the domains to the figure legend and the table. We have also changed the residue references throughout the paper to reflect the full-length protein to remain consistent. We also updated Sfig 1 to include color coding of the residues that correspond to the colors in fig 1 identifying the different subdomains in the structure.

73. Fig 3. zPol θ binds tightly to ds DNA – Line 151 says that “products were separated on a 6% non-denaturing gel”, however, line 146 contradicts saying “products were separated on a denaturing gel” (probably incorrect, since line 490 states “Native PAGE”). Furthermore, Fig. 3 shows only the quantitation, not the gel. Since the authors declare in Data Availability that “All relevant data are within the manuscript and its Supporting Information files”, gels should be shown too, either in the main Figure 3, or in the Supplementary file. Moreover, since the experiment should be performed three times, error bars should be included in the graph. – The text was corrected for the typo to read “non-denaturing gel”. The error bars were added to the figure and a representative gel was added to the raw data file.

4. Fig 4. zPol θ experiences greater nucleotide extension activity compared to hPol θ. - Line 182 states that “Under steady-state conditions 50 nM hPol θ or zPol θ were preincubated with 200 nM 25/40 dsDNA, however line 159 says (probably incorrectly) that “Under standard steady-state conditions, 200 nM of zPol θ or hPol θ was preincubated with 50 nM 25/40 dsDNA”. We recognize that this is inconsistent. Figure 4 reflects steady-state conditions and have rectified the error in the text.

5. Fig 5. zPol θ experiences biphasic burst activity. - Again, like for Fig. 3, it would be nice to see the gels that were quantitated here. – the raw gel images were added to the raw data file.

6. Fig. 6 zPol θ is able to perform MMEJ activity - This figure is not perspicuous. Firstly, probably a C is missing from the lower strand, and this way it looks as if there is a G:G mispairing with the upper strand. Secondly, it is not clear how many nucleotides Pol θ adds to the ssDNA. Explanations showing ’n’ and ’n+x’ are missing (In the raw gels file there is a caption “double-stranded DNA”). Interestingly, it looks as if a lot of nucleotides are added, at least 10, even though the templating part in the structure is only 6 nt long. Or is the run differe

---

## [Editor Report · Decision Letter 1]

10 Feb 2025

PONE-D-24-49625R1Zebrafish Polymerase Theta and human Polymerase Theta: orthologues with homologous function.PLOS ONE

Dear Dr. Weicksel,

Thank you for submitting your manuscript to PLOS ONE. After careful consideration, we feel that it has merit but does not fully meet PLOS ONE’s publication criteria as it currently stands. Therefore, we invite you to submit a revised version of the manuscript that addresses the points raised during the review process. Tiff file of Figure 3 seems missing. When we click button for Figure 3, it downloads figure 4 and so on... Please correct it, recheck it carefully before resubmitting.

 Please submit your revised manuscript by Mar 27 2025 11:59PM. If you will need more time than this to complete your revisions, please reply to this message or contact the journal office at plosone@plos.org . Please include the following items when submitting your revised manuscript:

We look forward to receiving your revised manuscript.

Kind regards,

Hari S. Misra, Ph.D.

Academic Editor

PLOS ONE
---

## [Editor Report · Decision Letter 2]

12 Mar 2025

Zebrafish Polymerase Theta and human Polymerase Theta: orthologues with homologous function.

PONE-D-24-49625R2

Dear Dr. Weicksel,

We’re pleased to inform you that your manuscript has been judged scientifically suitable for publication and will be formally accepted for publication once it meets all outstanding technical requirements.

Kind regards,

Hari S. Misra, Ph.D.

Academic Editor

PLOS ONE
---

## [Editor Report · Acceptance letter]

PONE-D-24-49625R2

PLOS ONE

Dear Dr. Weicksel,

I'm pleased to inform you that your manuscript has been deemed suitable for publication in PLOS ONE. Congratulations! Your manuscript is now being handed over to our production team.

Kind regards,

on behalf of

Professor Hari S. Misra

Academic Editor

PLOS ONE